# Generative Adversarial Imitation Learning

**Jonathan Ho**
OpenAI
hoj@openai.com

**Stefano Ermon**
Stanford University
ermon@cs.stanford.edu

## Abstract

Consider learning a policy from example expert behavior, without interaction with the expert or access to a reinforcement signal. One approach is to recover the expert's cost function with inverse reinforcement learning, then extract a policy from that cost function with reinforcement learning. This approach is indirect and can be slow. We propose a new general framework for directly extracting a policy from data as if it were obtained by reinforcement learning following inverse reinforcement learning. We show that a certain instantiation of our framework draws an analogy between imitation learning and generative adversarial networks, from which we derive a model-free imitation learning algorithm that obtains significant performance gains over existing model-free methods in imitating complex behaviors in large, high-dimensional environments.

## 1 Introduction

We are interested in a specific setting of imitation learning—the problem of learning to perform a task from expert demonstrations—in which the learner is given only samples of trajectories from the expert, is not allowed to query the expert for more data while training, and is not provided a reinforcement signal of any kind. There are two main approaches suitable for this setting: behavioral cloning [18], which learns a policy as a supervised learning problem over state-action pairs from expert trajectories; and inverse reinforcement learning [23, 16], which finds a cost function under which the expert is uniquely optimal.

Behavioral cloning, while appealingly simple, only tends to succeed with large amounts of data, due to compounding error caused by covariate shift [21, 22]. Inverse reinforcement learning (IRL), on the other hand, learns a cost function that prioritizes entire trajectories over others, so compounding error, a problem for methods that fit single-timestep decisions, is not an issue. Accordingly, IRL has succeeded in a wide range of problems, from predicting behaviors of taxi drivers [29] to planning footsteps for quadruped robots [20].

Unfortunately, many IRL algorithms are extremely expensive to run, requiring reinforcement learning in an inner loop. Scaling IRL methods to large environments has thus been the focus of much recent work [6, 13]. Fundamentally, however, IRL learns a cost function, which explains expert behavior but does not directly tell the learner how to act. Given that the learner's true goal often is to take actions imitating the expert—indeed, many IRL algorithms are evaluated on the quality of the optimal actions of the costs they learn—why, then, must we learn a cost function, if doing so possibly incurs significant computational expense yet fails to directly yield actions?

We desire an algorithm that tells us explicitly how to act by directly learning a policy. To develop such an algorithm, we begin in Section 3, where we characterize the policy given by running reinforcement learning on a cost function learned by maximum causal entropy IRL [29, 30]. Our characterization introduces a framework for directly learning policies from data, bypassing any intermediate IRL step.

Then, we instantiate our framework in Sections 4 and 5 with a new model-free imitation learning algorithm. We show that our resulting algorithm is intimately connected to generative adversarial

networks [8], a technique from the deep learning community that has led to recent successes in modeling distributions of natural images: our algorithm harnesses generative adversarial training to fit distributions of states and actions defining expert behavior. We test our algorithm in Section 6, where we find that it outperforms competing methods by a wide margin in training policies for complex, high-dimensional physics-based control tasks over various amounts of expert data.

## 2 Background

**Preliminaries**   $\overline{\mathbb{R}}$ will denote the extended real numbers $\mathbb{R} \cup \{\infty\}$. Section 3 will work with finite state and action spaces $\mathcal{S}$ and $\mathcal{A}$, but our algorithms and experiments later in the paper will run in high-dimensional continuous environments. $\Pi$ is the set of all stationary stochastic policies that take actions in $\mathcal{A}$ given states in $\mathcal{S}$; successor states are drawn from the dynamics model $P(s'|s,a)$. We work in the $\gamma$-discounted infinite horizon setting, and we will use an expectation with respect a policy $\pi \in \Pi$ to denote an expectation with respect to the trajectory it generates: $\mathbb{E}_\pi[c(s,a)] \triangleq \mathbb{E}\left[\sum_{t=0}^\infty \gamma^t c(s_t, a_t)\right]$, where $s_0 \sim p_0$, $a_t \sim \pi(\cdot|s_t)$, and $s_{t+1} \sim P(\cdot|s_t, a_t)$ for $t \geq 0$. We will use $\hat{\mathbb{E}}_\tau$ to denote an empirical expectation with respect to trajectory samples $\tau$, and we will always use $\pi_E$ to refer to the expert policy.

**Inverse reinforcement learning**   Suppose we are given an expert policy $\pi_E$ that we wish to rationalize with IRL. For the remainder of this paper, we will adopt and assume the existence of solutions of maximum causal entropy IRL [29, 30], which fits a cost function from a family of functions $\mathcal{C}$ with the optimization problem

$$\underset{c \in \mathcal{C}}{\text{maximize}} \left( \min_{\pi \in \Pi} -H(\pi) + \mathbb{E}_\pi[c(s,a)] \right) - \mathbb{E}_{\pi_E}[c(s,a)] \tag{1}$$

where $H(\pi) \triangleq \mathbb{E}_\pi[-\log \pi(a|s)]$ is the $\gamma$-discounted causal entropy [3] of the policy $\pi$. In practice, $\pi_E$ will only be provided as a set of trajectories sampled by executing $\pi_E$ in the environment, so the expected cost of $\pi_E$ in Eq. (1) is estimated using these samples. Maximum causal entropy IRL looks for a cost function $c \in \mathcal{C}$ that assigns low cost to the expert policy and high cost to other policies, thereby allowing the expert policy to be found via a certain reinforcement learning procedure:

$$\text{RL}(c) = \underset{\pi \in \Pi}{\arg\min} -H(\pi) + \mathbb{E}_\pi[c(s,a)] \tag{2}$$

which maps a cost function to high-entropy policies that minimize the expected cumulative cost.

## 3 Characterizing the induced optimal policy

To begin our search for an imitation learning algorithm that both bypasses an intermediate IRL step and is suitable for large environments, we will study policies found by reinforcement learning on costs learned by IRL on the largest possible set of cost functions $\mathcal{C}$ in Eq. (1): *all* functions $\mathbb{R}^{\mathcal{S} \times \mathcal{A}} = \{c : \mathcal{S} \times \mathcal{A} \to \mathbb{R}\}$. Using expressive cost function classes, like Gaussian processes [14] and neural networks [6], is crucial to properly explain complex expert behavior without meticulously hand-crafted features. Here, we investigate the best IRL can do with respect to expressiveness by examining its capabilities with $\mathcal{C} = \mathbb{R}^{\mathcal{S} \times \mathcal{A}}$.

Of course, with such a large $\mathcal{C}$, IRL can easily overfit when provided a finite dataset. Therefore, we will incorporate a (closed, proper) convex cost function regularizer $\psi : \mathbb{R}^{\mathcal{S} \times \mathcal{A}} \to \overline{\mathbb{R}}$ into our study. Note that convexity is a not particularly restrictive requirement: $\psi$ must be convex as a function defined on all of $\mathbb{R}^{\mathcal{S} \times \mathcal{A}}$, not as a function defined on a small parameter space; indeed, the cost regularizers of Finn et al. [6], effective for a range of robotic manipulation tasks, satisfy this requirement. Interestingly, $\psi$ will play a central role in our discussion and will not serve as a nuisance in our analysis.

Let us define an IRL primitive procedure, which finds a cost function such that the expert performs better than all other policies, with the cost regularized by $\psi$:

$$\text{IRL}_\psi(\pi_E) = \underset{c \in \mathbb{R}^{\mathcal{S} \times \mathcal{A}}}{\arg\max} -\psi(c) + \left( \min_{\pi \in \Pi} -H(\pi) + \mathbb{E}_\pi[c(s,a)] \right) - \mathbb{E}_{\pi_E}[c(s,a)] \tag{3}$$

Let $\tilde{c} \in \mathrm{IRL}_\psi(\pi_E)$. We are interested in a policy given by $\mathrm{RL}(\tilde{c})$—this is the policy given by running reinforcement learning on the output of IRL. To characterize $\mathrm{RL}(\tilde{c})$, let us first define for a policy $\pi \in \Pi$ its occupancy measure $\rho_\pi : \mathcal{S} \times \mathcal{A} \to \mathbb{R}$ as $\rho_\pi(s, a) = \pi(a|s) \sum_{t=0}^\infty \gamma^t P(s_t = s|\pi)$. The occupancy measure can be interpreted as the unnormalized distribution of state-action pairs that an agent encounters when navigating the environment with the policy $\pi$, and it allows us to write $\mathbb{E}_\pi[c(s, a)] = \sum_{s,a} \rho_\pi(s, a)c(s, a)$ for any cost function $c$. We will also need the concept of a convex conjugate: for a function $f : \mathbb{R}^{\mathcal{S} \times \mathcal{A}} \to \overline{\mathbb{R}}$, its convex conjugate $f^* : \mathbb{R}^{\mathcal{S} \times \mathcal{A}} \to \overline{\mathbb{R}}$ is given by $f^*(x) = \sup_{y \in \mathbb{R}^{\mathcal{S} \times \mathcal{A}}} x^T y - f(y)$.

Now, we are prepared to characterize $\mathrm{RL}(\tilde{c})$, the policy learned by RL on the cost recovered by IRL:

**Proposition 3.1.** $\mathrm{RL} \circ \mathrm{IRL}_\psi(\pi_E) = \arg\min_{\pi \in \Pi} -H(\pi) + \psi^*(\rho_\pi - \rho_{\pi_E})$ (4)

The proof of Proposition 3.1 can be found in Appendix A.1. It relies on the observation that the optimal cost function and policy form a saddle point of a certain function. IRL finds one coordinate of this saddle point, and running RL on the output of IRL reveals the other coordinate.

Proposition 3.1 tells us that $\psi$-regularized inverse reinforcement learning, implicitly, seeks a policy whose occupancy measure is close to the expert's, as measured by $\psi^*$. Enticingly, this suggests that various settings of $\psi$ lead to various imitation learning algorithms that directly solve the optimization problem given by Proposition 3.1. We explore such algorithms in Sections 4 and 5, where we show that certain settings of $\psi$ lead to both existing algorithms and a novel one.

The special case when $\psi$ is a constant function is particularly illuminating, so we state and show it directly using concepts from convex optimization.

**Proposition 3.2.** *Suppose $\rho_{\pi_E} > 0$. If $\psi$ is a constant function, $\tilde{c} \in \mathrm{IRL}_\psi(\pi_E)$, and $\tilde{\pi} \in \mathrm{RL}(\tilde{c})$, then $\rho_{\tilde{\pi}} = \rho_{\pi_E}$.*

In other words, if there were no cost regularization at all, the recovered policy will exactly match the expert's occupancy measure. (The condition $\rho_{\pi_E} > 0$, inherited from Ziebart et al. [30], simplifies our discussion and in fact guarantees the existence of $\tilde{c} \in \mathrm{IRL}_\psi(\pi_E)$. Elsewhere in the paper, as mentioned in Section 2, we assume the IRL problem has a solution.) To show Proposition 3.2, we need the basic result that the set of valid occupancy measures $\mathcal{D} \triangleq \{\rho_\pi : \pi \in \Pi\}$ can be written as a feasible set of affine constraints [19]: if $p_0(s)$ is the distribution of starting states and $P(s'|s, a)$ is the dynamics model, then $\mathcal{D} = \left\{ \rho : \rho \geq 0 \quad \text{and} \quad \sum_a \rho(s, a) = p_0(s) + \gamma \sum_{s',a} P(s|s', a)\rho(s', a) \ \forall s \in \mathcal{S} \right\}$. Furthermore, there is a one-to-one correspondence between $\Pi$ and $\mathcal{D}$:

**Lemma 3.1** (Theorem 2 of Syed et al. [27]). *If $\rho \in \mathcal{D}$, then $\rho$ is the occupancy measure for $\pi_\rho(a|s) \triangleq \rho(s, a)/ \sum_{a'} \rho(s, a')$, and $\pi_\rho$ is the only policy whose occupancy measure is $\rho$.*

We are therefore justified in writing $\pi_\rho$ to denote the unique policy for an occupancy measure $\rho$. We also need a lemma that lets us speak about causal entropies of occupancy measures:

**Lemma 3.2.** *Let $\bar{H}(\rho) = -\sum_{s,a} \rho(s, a) \log(\rho(s, a)/ \sum_{a'} \rho(s, a'))$. Then, $\bar{H}$ is strictly concave, and for all $\pi \in \Pi$ and $\rho \in \mathcal{D}$, we have $H(\pi) = \bar{H}(\rho_\pi)$ and $\bar{H}(\rho) = H(\pi_\rho)$.*

The proof of this lemma is in Appendix A.1. Lemma 3.1 and Lemma 3.2 together allow us to freely switch between policies and occupancy measures when considering functions involving causal entropy and expected costs, as in the following lemma:

**Lemma 3.3.** *If $L(\pi, c) = -H(\pi) + \mathbb{E}_\pi[c(s, a)]$ and $\bar{L}(\rho, c) = -\bar{H}(\rho) + \sum_{s,a} \rho(s, a)c(s, a)$, then, for all cost functions $c$, $L(\pi, c) = \bar{L}(\rho_\pi, c)$ for all policies $\pi \in \Pi$, and $\bar{L}(\rho, c) = L(\pi_\rho, c)$ for all occupancy measures $\rho \in \mathcal{D}$.*

Now, we are ready to verify Proposition 3.2.

*Proof of Proposition 3.2.* Define $\bar{L}(\rho, c) = -\bar{H}(\rho) + \sum_{s,a} c(s, a)(\rho(s, a) - \rho_E(s, a))$. Given that $\psi$ is a constant function, we have the following, due to Lemma 3.3:

$$\tilde{c} \in \mathrm{IRL}_\psi(\pi_E) = \arg\max_{c \in \mathbb{R}^{\mathcal{S} \times \mathcal{A}}} \min_{\pi \in \Pi} -H(\pi) + \mathbb{E}_\pi[c(s, a)] - \mathbb{E}_{\pi_E}[c(s, a)] + \mathrm{const}.$$ (5)

$$= \arg\max_{c \in \mathbb{R}^{\mathcal{S} \times \mathcal{A}}} \min_{\rho \in \mathcal{D}} -\bar{H}(\rho) + \sum_{s,a} \rho(s, a)c(s, a) - \sum_{s,a} \rho_E(s, a)c(s, a) = \arg\max_{c \in \mathbb{R}^{\mathcal{S} \times \mathcal{A}}} \min_{\rho \in \mathcal{D}} \bar{L}(\rho, c).$$ (6)

This is the dual of the optimization problem

$$\underset{\rho \in \mathcal{D}}{\text{minimize}} -\bar{H}(\rho) \quad \text{subject to} \quad \rho(s, a) = \rho_E(s, a) \quad \forall\, s \in \mathcal{S}, a \in \mathcal{A} \qquad (7)$$

with Lagrangian $\bar{L}$, for which the costs $c(s, a)$ serve as dual variables for equality constraints. Thus, $\tilde{c}$ is a dual optimum for (7). In addition, strong duality holds for (7): $\mathcal{D}$ is compact and convex, $-\bar{H}$ is convex, and, since $\rho_E > 0$, there exists a feasible point in the relative interior of the domain $\mathcal{D}$. Moreover, Lemma 3.2 guarantees that $-\bar{H}$ is in fact strictly convex, so the primal optimum can be uniquely recovered from the dual optimum [4, Section 5.5.5] via $\tilde{\rho} = \arg\min_{\rho \in \mathcal{D}} \bar{L}(\rho, \tilde{c}) = \arg\min_{\rho \in \mathcal{D}} -\bar{H}(\rho) + \sum_{s,a} \tilde{c}(s, a)\rho(s, a) = \rho_E$, where the first equality indicates that $\tilde{\rho}$ is the unique minimizer of $\bar{L}(\cdot, \tilde{c})$, and the third follows from the constraints in the primal problem (7). But if $\tilde{\pi} \in \text{RL}(\tilde{c})$, then Lemma 3.3 implies $\rho_{\tilde{\pi}} = \tilde{\rho} = \rho_E$. □

Let us summarize our conclusions. First, **IRL is a dual of an occupancy measure matching problem**, and the recovered cost function is the dual optimum. Classic IRL algorithms that solve reinforcement learning repeatedly in an inner loop, such as the algorithm of Ziebart et al. [29] that runs a variant of value iteration in an inner loop, can be interpreted as a form of dual ascent, in which one repeatedly solves the primal problem (reinforcement learning) with fixed dual values (costs). Dual ascent is effective if solving the unconstrained primal is efficient, but in the case of IRL, it amounts to reinforcement learning! Second, **the induced optimal policy is the primal optimum.** The induced optimal policy is obtained by running RL after IRL, which is exactly the act of recovering the primal optimum from the dual optimum; that is, optimizing the Lagrangian with the dual variables fixed at the dual optimum values. Strong duality implies that this induced optimal policy is indeed the primal optimum, and therefore matches occupancy measures with the expert. IRL is traditionally defined as the act of finding a cost function such that the expert policy is uniquely optimal, but we can alternatively view IRL as a procedure that tries to *induce a policy that matches the expert's occupancy measure*.

## 4   Practical occupancy measure matching

We saw in Proposition 3.2 that if $\psi$ is constant, the resulting primal problem (7) simply matches occupancy measures with expert at all states and actions. Such an algorithm is not practically useful. In reality, the expert trajectory distribution will be provided only as a finite set of samples, so in large environments, most of the expert's occupancy measure values will be small, and exact occupancy measure matching will force the learned policy to rarely visit these unseen state-action pairs simply due to lack of data. Furthermore, in the cases in which we would like to use function approximation to learn parameterized policies $\pi_\theta$, the resulting optimization problem of finding an appropriate $\theta$ would have an intractably large number of constraints when the environment is large: as many constraints as points in $\mathcal{S} \times \mathcal{A}$.

Keeping in mind that we wish to eventually develop an imitation learning algorithm suitable for large environments, we would like to relax Eq. (7) into the following form, motivated by Proposition 3.1:

$$\underset{\pi}{\text{minimize}} \; d_\psi(\rho_\pi, \rho_E) - H(\pi) \qquad (8)$$

by modifying the IRL regularizer $\psi$ so that $d_\psi(\rho_\pi, \rho_E) \triangleq \psi^*(\rho_\pi - \rho_E)$ smoothly penalizes violations in difference between the occupancy measures.

**Entropy-regularized apprenticeship learning**   It turns out that with certain settings of $\psi$, Eq. (8) takes on the form of regularized variants of existing *apprenticeship learning* algorithms, which indeed do scale to large environments with parameterized policies [10]. For a class of cost functions $\mathcal{C} \subset \mathbb{R}^{\mathcal{S} \times \mathcal{A}}$, an apprenticeship learning algorithm finds a policy that performs better than the expert across $\mathcal{C}$, by optimizing the objective

$$\underset{\pi}{\text{minimize}} \; \underset{c \in \mathcal{C}}{\max} \; \mathbb{E}_\pi[c(s, a)] - \mathbb{E}_{\pi_E}[c(s, a)] \qquad (9)$$

Classic apprenticeship learning algorithms restrict $\mathcal{C}$ to convex sets given by linear combinations of basis functions $f_1, \ldots, f_d$, which give rise a feature vector $f(s, a) = [f_1(s, a), \ldots, f_d(s, a)]$ for each state-action pair. Abbeel and Ng [1] and Syed et al. [27] use, respectively,

$$\mathcal{C}_{\text{linear}} = \{\textstyle\sum_i w_i f_i \; : \; \|w\|_2 \leq 1\} \quad \text{and} \quad \mathcal{C}_{\text{convex}} = \{\textstyle\sum_i w_i f_i \; : \; \textstyle\sum_i w_i = 1, w_i \geq 0 \;\forall i\}. \qquad (10)$$

$\mathcal{C}_{\text{linear}}$ leads to feature expectation matching [1], which minimizes $\ell_2$ distance between expected feature vectors: $\max_{c \in \mathcal{C}_{\text{linear}}} \mathbb{E}_\pi[c(s,a)] - \mathbb{E}_{\pi_E}[c(s,a)] = \|\mathbb{E}_\pi[f(s,a)] - \mathbb{E}_{\pi_E}[f(s,a)]\|_2$. Meanwhile, $\mathcal{C}_{\text{convex}}$ leads to MWAL [26] and LPAL [27], which minimize worst-case excess cost among the individual basis functions, as $\max_{c \in \mathcal{C}_{\text{convex}}} \mathbb{E}_\pi[c(s,a)] - \mathbb{E}_{\pi_E}[c(s,a)] = \max_{i \in \{1,\ldots,d\}} \mathbb{E}_\pi[f_i(s,a)] - \mathbb{E}_{\pi_E}[f_i(s,a)]$.

We now show how Eq. (9) is a special case of Eq. (8) with a certain setting of $\psi$. With the indicator function $\delta_{\mathcal{C}} : \mathbb{R}^{S \times A} \to \overline{\mathbb{R}}$, defined by $\delta_{\mathcal{C}}(c) = 0$ if $c \in \mathcal{C}$ and $+\infty$ otherwise, we can write the apprenticeship learning objective (9) as

$$\max_{c \in \mathcal{C}} \mathbb{E}_\pi[c(s,a)] - \mathbb{E}_{\pi_E}[c(s,a)] = \max_{c \in \mathbb{R}^{S \times A}} -\delta_{\mathcal{C}}(c) + \sum_{s,a} (\rho_\pi(s,a) - \rho_{\pi_E}(s,a))c(s,a) = \delta_{\mathcal{C}}^*(\rho_\pi - \rho_{\pi_E})$$

Therefore, we see that entropy-regularized apprenticeship learning

$$\underset{\pi}{\text{minimize}} -H(\pi) + \max_{c \in \mathcal{C}} \mathbb{E}_\pi[c(s,a)] - \mathbb{E}_{\pi_E}[c(s,a)] \tag{11}$$

is equivalent to performing RL following IRL with cost regularizer $\psi = \delta_{\mathcal{C}}$, which forces the implicit IRL procedure to recover a cost function lying in $\mathcal{C}$. Note that we can scale the policy's entropy regularization strength in Eq. (11) by scaling $\mathcal{C}$ by a constant $\alpha$ as $\{\alpha c : c \in \mathcal{C}\}$, recovering the original apprenticeship objective (9) by taking $\alpha \to \infty$.

**Cons of apprenticeship learning** It is known that apprenticeship learning algorithms generally do not recover expert-like policies if $\mathcal{C}$ is too restrictive [27, Section 1]—which is often the case for the linear subspaces used by feature expectation matching, MWAL, and LPAL, unless the basis functions $f_1, \ldots, f_d$ are very carefully designed. Intuitively, unless the true expert cost function (assuming it exists) lies in $\mathcal{C}$, there is no guarantee that if $\pi$ performs better than $\pi_E$ on all of $\mathcal{C}$, then $\pi$ equals $\pi_E$. With the aforementioned insight based on Proposition 3.1 that apprenticeship learning is equivalent to RL following IRL, we can understand exactly why apprenticeship learning may fail to imitate: it forces $\pi_E$ to be encoded as an element of $\mathcal{C}$. If $\mathcal{C}$ does not include a cost function that explains expert behavior well, then attempting to recover a policy from such an encoding will not succeed.

**Pros of apprenticeship learning** While restrictive cost classes $\mathcal{C}$ may not lead to exact imitation, apprenticeship learning with such $\mathcal{C}$ can scale to large state and action spaces with policy function approximation. Ho et al. [10] rely on the following policy gradient formula for the apprenticeship objective (9) for a parameterized policy $\pi_\theta$:

$$\nabla_\theta \max_{c \in \mathcal{C}} \mathbb{E}_{\pi_\theta}[c(s,a)] - \mathbb{E}_{\pi_E}[c(s,a)] = \nabla_\theta \mathbb{E}_{\pi_\theta}[c^*(s,a)] = \mathbb{E}_{\pi_\theta}[\nabla_\theta \log \pi_\theta(a|s)Q_{c^*}(s,a)]$$

$$\text{where } c^* = \underset{c \in \mathcal{C}}{\arg\max} \, \mathbb{E}_{\pi_\theta}[c(s,a)] - \mathbb{E}_{\pi_E}[c(s,a)], \; Q_{c^*}(\bar{s}, \bar{a}) = \mathbb{E}_{\pi_\theta}[c^*(\bar{s}, \bar{a}) \mid s_0 = \bar{s}, a_0 = \bar{a}] \tag{12}$$

Observing that Eq. (12) is the policy gradient for a reinforcement learning objective with cost $c^*$, Ho et al. propose an algorithm that alternates between two steps:

1. Sample trajectories of the current policy $\pi_{\theta_i}$ by simulating in the environment, and fit a cost function $c_i^*$, as defined in Eq. (12). For the cost classes $\mathcal{C}_{\text{linear}}$ and $\mathcal{C}_{\text{convex}}$ (10), this cost fitting amounts to evaluating simple analytical expressions [10].

2. Form a gradient estimate with Eq. (12) with $c_i^*$ and the sampled trajectories, and take a trust region policy optimization (TRPO) [24] step to produce $\pi_{\theta_{i+1}}$.

This algorithm relies crucially on the TRPO policy step, which is a natural gradient step constrained to ensure that $\pi_{\theta_{i+1}}$ does not stray too far $\pi_{\theta_i}$, as measured by KL divergence between the two policies averaged over the states in the sampled trajectories. This carefully constructed step scheme ensures that the algorithm does not diverge due to high noise in estimating the gradient (12). We refer the reader to Schulman et al. [24] for more details on TRPO.

With the TRPO step scheme, Ho et al. were able train large neural network policies for apprenticeship learning with linear cost function classes (10) in environments with hundreds of observation dimensions. Their use of these linear cost function classes, however, limits their approach to settings in which expert behavior is well-described by such classes. We will draw upon their algorithm to develop an imitation learning method that both scales to large environments and imitates arbitrarily complex expert behavior. To do so, we first turn to proposing a new regularizer $\psi$ that wields more expressive power than the regularizers corresponding to $\mathcal{C}_{\text{linear}}$ and $\mathcal{C}_{\text{convex}}$ (10).

# 5 Generative adversarial imitation learning

As discussed in Section 4, the constant regularizer leads to an imitation learning algorithm that exactly matches occupancy measures, but is intractable in large environments. The indicator regularizers for the linear cost function classes (10), on the other hand, lead to algorithms incapable of exactly matching occupancy measures without careful tuning, but are tractable in large environments. We propose the following new cost regularizer that combines the best of both worlds, as we will show in the coming sections:

$$\psi_{\text{GA}}(c) \triangleq \begin{cases} \mathbb{E}_{\pi_E}[g(c(s,a))] & \text{if } c < 0 \\ +\infty & \text{otherwise} \end{cases} \quad \text{where } g(x) = \begin{cases} -x - \log(1 - e^x) & \text{if } x < 0 \\ +\infty & \text{otherwise} \end{cases} \quad (13)$$

This regularizer places low penalty on cost functions $c$ that assign an amount of negative cost to expert state-action pairs; if $c$, however, assigns large costs (close to zero, which is the upper bound for costs feasible for $\psi_{\text{GA}}$) to the expert, then $\psi_{\text{GA}}$ will heavily penalize $c$. An interesting property of $\psi_{\text{GA}}$ is that it is an average over expert data, and therefore can adjust to arbitrary expert datasets. The indicator regularizers $\delta_{\mathcal{C}}$, used by the linear apprenticeship learning algorithms described in Section 4, are always fixed, and cannot adapt to data as $\psi_{\text{GA}}$ can. Perhaps the most important difference between $\psi_{\text{GA}}$ and $\delta_{\mathcal{C}}$, however, is that $\delta_{\mathcal{C}}$ forces costs to lie in a small subspace spanned by finitely many basis functions, whereas $\psi_{\text{GA}}$ allows for any cost function, as long as it is negative everywhere.

Our choice of $\psi_{\text{GA}}$ is motivated by the following fact, shown in the appendix (Corollary A.1.1):

$$\psi_{\text{GA}}^*(\rho_\pi - \rho_{\pi_E}) = \sup_{D \in (0,1)^{\mathcal{S} \times \mathcal{A}}} \mathbb{E}_\pi[\log(D(s,a))] + \mathbb{E}_{\pi_E}[\log(1 - D(s,a))] \quad (14)$$

where the supremum ranges over discriminative classifiers $D : \mathcal{S} \times \mathcal{A} \to (0,1)$. Equation (14) is proportional to the optimal negative log loss of the binary classification problem of distinguishing between state-action pairs of $\pi$ and $\pi_E$. It turns out that this optimal loss is, up to a constant shift and scaling, the Jensen-Shannon divergence $D_{\text{JS}}(\bar{\rho}_\pi, \bar{\rho}_{\pi_E}) \triangleq D_{\text{KL}}(\bar{\rho}_\pi \| (\bar{\rho}_\pi + \bar{\rho}_E)/2) + D_{\text{KL}}(\bar{\rho}_E \| (\bar{\rho}_\pi + \bar{\rho}_E)/2)$, which is a squared metric between the normalized occupancy distributions $\bar{\rho}_\pi = (1 - \gamma)\rho_\pi$ and $\bar{\rho}_{\pi_E} = (1 - \gamma)\rho_{\pi_E}$ [8, 17]. Treating the causal entropy $H$ as a policy regularizer controlled by $\lambda \geq 0$ and dropping the $1 - \gamma$ occupancy measure normalization for clarity, we obtain a new imitation learning algorithm:

$$\underset{\pi}{\text{minimize}} \ \psi_{\text{GA}}^*(\rho_\pi - \rho_{\pi_E}) - \lambda H(\pi) = D_{\text{JS}}(\rho_\pi, \rho_{\pi_E}) - \lambda H(\pi), \quad (15)$$

which finds a policy whose occupancy measure minimizes Jensen-Shannon divergence to the expert's. Equation (15) minimizes a true metric between occupancy measures, so, unlike linear apprenticeship learning algorithms, it can imitate expert policies exactly.

**Algorithm** Equation (15) draws a connection between imitation learning and generative adversarial networks [8], which train a generative model $G$ by having it confuse a discriminative classifier $D$. The job of $D$ is to distinguish between the distribution of data generated by $G$ and the true data distribution. When $D$ cannot distinguish data generated by $G$ from the true data, then $G$ has successfully matched the true data. In our setting, the learner's occupancy measure $\rho_\pi$ is analogous to the data distribution generated by $G$, and the expert's occupancy measure $\rho_{\pi_E}$ is analogous to the true data distribution.

We now present a practical imitation learning algorithm, called *generative adversarial imitation learning* or GAIL (Algorithm 1), designed to work in large environments. GAIL solves Eq. (15) by finding a saddle point $(\pi, D)$ of the expression

$$\mathbb{E}_\pi[\log(D(s,a))] + \mathbb{E}_{\pi_E}[\log(1 - D(s,a))] - \lambda H(\pi) \quad (16)$$

with both $\pi$ and $D$ represented using function approximators: GAIL fits a parameterized policy $\pi_\theta$, with weights $\theta$, and a discriminator network $D_w : \mathcal{S} \times \mathcal{A} \to (0,1)$, with weights $w$. GAIL alternates between an Adam [11] gradient step on $w$ to increase Eq. (16) with respect to $D$, and a TRPO step on $\theta$ to decrease Eq. (16) with respect to $\pi$ (we derive an estimator for the causal entropy gradient $\nabla_\theta H(\pi_\theta)$ in Appendix A.2). The TRPO step serves the same purpose as it does with the apprenticeship learning algorithm of Ho et al. [10]: it prevents the policy from changing too much due to noise in the policy gradient. The discriminator network can be interpreted as a local cost function providing learning signal to the policy—specifically, taking a policy step that decreases expected cost with respect to the cost function $c(s,a) = \log D(s,a)$ will move toward expert-like regions of state-action space, as classified by the discriminator.

---
**Algorithm 1** Generative adversarial imitation learning
---
1: **Input:** Expert trajectories $\tau_E \sim \pi_E$, initial policy and discriminator parameters $\theta_0, w_0$
2: **for** $i = 0, 1, 2, \ldots$ **do**
3:     Sample trajectories $\tau_i \sim \pi_{\theta_i}$
4:     Update the discriminator parameters from $w_i$ to $w_{i+1}$ with the gradient

$$\hat{\mathbb{E}}_{\tau_i}[\nabla_w \log(D_w(s,a))] + \hat{\mathbb{E}}_{\tau_E}[\nabla_w \log(1 - D_w(s,a))] \qquad (17)$$

5:     Take a policy step from $\theta_i$ to $\theta_{i+1}$, using the TRPO rule with cost function $\log(D_{w_{i+1}}(s,a))$. Specifically, take a KL-constrained natural gradient step with

$$\hat{\mathbb{E}}_{\tau_i}[\nabla_\theta \log \pi_\theta(a|s) Q(s,a)] - \lambda \nabla_\theta H(\pi_\theta),$$
$$\text{where } Q(\bar{s}, \bar{a}) = \hat{\mathbb{E}}_{\tau_i}[\log(D_{w_{i+1}}(s,a)) \,|\, s_0 = \bar{s}, a_0 = \bar{a}] \qquad (18)$$

6: **end for**
---

# 6 Experiments

We evaluated GAIL against baselines on 9 physics-based control tasks, ranging from low-dimensional control tasks from the classic RL literature—the cartpole [2], acrobot [7], and mountain car [15]—to difficult high-dimensional tasks such as a 3D humanoid locomotion, solved only recently by model-free reinforcement learning [25, 24]. All environments, other than the classic control tasks, were simulated with MuJoCo [28]. See Appendix B for a complete description of all the tasks.

Each task comes with a true cost function, defined in the OpenAI Gym [5]. We first generated expert behavior for these tasks by running TRPO [24] on these true cost functions to create expert policies. Then, to evaluate imitation performance with respect to sample complexity of expert data, we sampled datasets of varying trajectory counts from the expert policies. The trajectories constituting each dataset each consisted of about 50 state-action pairs. We tested GAIL against three baselines:

1. Behavioral cloning: a given dataset of state-action pairs is split into 70% training data and 30% validation data. The policy is trained with supervised learning, using Adam [11] with minibatches of 128 examples, until validation error stops decreasing.

2. Feature expectation matching (FEM): the algorithm of Ho et al. [10] using the cost function class $\mathcal{C}_{\text{linear}}$ (10) of Abbeel and Ng [1]

3. Game-theoretic apprenticeship learning (GTAL): the algorithm of Ho et al. [10] using the cost function class $\mathcal{C}_{\text{convex}}$ (10) of Syed and Schapire [26]

We used all algorithms to train policies of the same neural network architecture for all tasks: two hidden layers of 100 units each, with tanh nonlinearities in between. The discriminator networks for GAIL also used the same architecture. All networks were always initialized randomly at the start of each trial. For each task, we gave FEM, GTAL, and GAIL exactly the same amount of environment interaction for training. We ran all algorithms 5-7 times over different random seeds in all environments except Humanoid, due to time restrictions.

Figure 1 depicts the results, and Appendix B provides exact performance numbers and details of our experiment pipeline, including expert data sampling and algorithm hyperparameters. We found that on the classic control tasks (cartpole, acrobot, and mountain car), behavioral cloning generally suffered in expert data efficiency compared to FEM and GTAL, which for the most part were able produce policies with near-expert performance with a wide range of dataset sizes, albeit with large variance over different random initializations of the policy. On these tasks, GAIL consistently produced policies performing better than behavioral cloning, FEM, and GTAL. However, behavioral cloning performed excellently on the Reacher task, on which it was more sample efficient than GAIL. We were able to slightly improve GAIL's performance on Reacher using causal entropy regularization—in the 4-trajectory setting, the improvement from $\lambda = 0$ to $\lambda = 10^{-3}$ was statistically significant over training reruns, according to a one-sided Wilcoxon rank-sum test with $p = .05$. We used no causal entropy regularization for all other tasks.

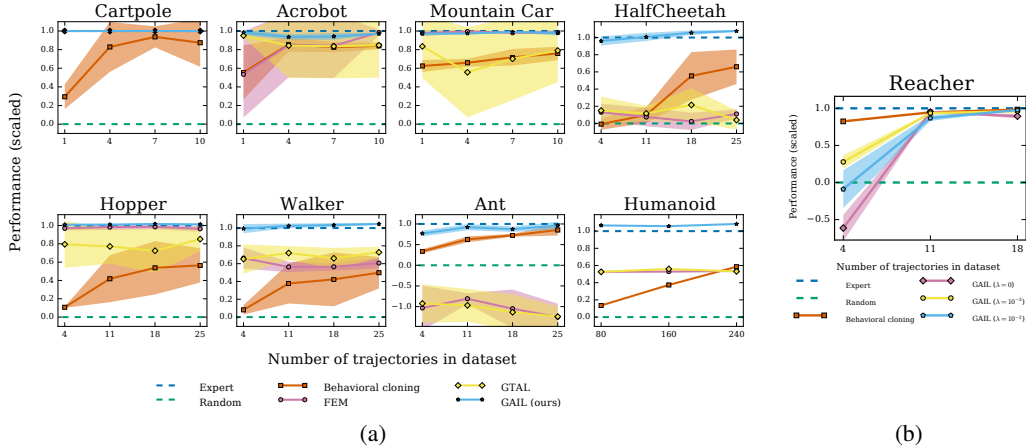

Figure 1: (a) Performance of learned policies. The $y$-axis is negative cost, scaled so that the expert achieves 1 and a random policy achieves 0. (b) Causal entropy regularization $\lambda$ on Reacher. Except for Humanoid, shading indicates standard deviation over 5-7 reruns.

On the other MuJoCo environments, GAIL almost always achieved at least 70% of expert performance for all dataset sizes we tested and reached it exactly with the larger datasets, with very little variance among random seeds. The baseline algorithms generally could not reach expert performance even with the largest datasets. FEM and GTAL performed poorly for Ant, producing policies consistently worse than a policy that chooses actions uniformly at random. Behavioral cloning was able to reach satisfactory performance with enough data on HalfCheetah, Hopper, Walker, and Ant, but was unable to achieve more than 60% for Humanoid, on which GAIL achieved exact expert performance for all tested dataset sizes.

# 7 Discussion and outlook

As we demonstrated, GAIL is generally quite sample efficient in terms of expert data. However, it is not particularly sample efficient in terms of environment interaction during training. The number of such samples required to estimate the imitation objective gradient (18) was comparable to the number needed for TRPO to train the expert policies from reinforcement signals. We believe that we could significantly improve learning speed for GAIL by initializing policy parameters with behavioral cloning, which requires no environment interaction at all.

Fundamentally, our method is model free, so it will generally need more environment interaction than model-based methods. Guided cost learning [6], for instance, builds upon guided policy search [12] and inherits its sample efficiency, but also inherits its requirement that the model is well-approximated by iteratively fitted time-varying linear dynamics. Interestingly, both GAIL and guided cost learning alternate between policy optimization steps and cost fitting (which we called discriminator fitting), even though the two algorithms are derived completely differently.

Our approach builds upon a vast line of work on IRL [29, 1, 27, 26], and hence, just like IRL, our approach does not interact with the expert during training. Our method explores randomly to determine which actions bring a policy's occupancy measure closer to the expert's, whereas methods that do interact with the expert, like DAgger [22], can simply ask the expert for such actions. Ultimately, we believe that a method that combines well-chosen environment models with expert interaction will win in terms of sample complexity of both expert data and environment interaction.

### Acknowledgments

We thank Jayesh K. Gupta, John Schulman, and the anonymous reviewers for assistance, advice, and critique. This work was supported by the SAIL-Toyota Center for AI Research and by a NSF Graduate Research Fellowship (grant no. DGE-114747).

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
