[Supplementary Material]

# A  Proofs

## A.1  Proofs for Section 3

*Proof of Lemma 3.2.*  First, we show strict concavity of $\bar{H}$. Let $\rho$ and $\rho'$ be occupancy measures, and suppose $\lambda \in [0, 1]$. For all $s$ and $a$, the log-sum inequality implies:

$$-(\lambda\rho(s,a) + (1-\lambda)\rho'(s,a)) \log \frac{\lambda\rho(s,a) + (1-\lambda)\rho'(s,a)}{\sum_{a'}(\lambda\rho(s,a') + (1-\lambda)\rho'(s,a'))} \tag{19}$$

$$= -(\lambda\rho(s,a) + (1-\lambda)\rho'(s,a)) \log \frac{\lambda\rho(s,a) + (1-\lambda)\rho'(s,a)}{\lambda\sum_{a'}\rho(s,a') + (1-\lambda)\sum_{a'}\rho'(s,a')} \tag{20}$$

$$\geq -\lambda\rho(s,a) \log \frac{\lambda\rho(s,a)}{\lambda\sum_{a'}\rho(s,a')} - (1-\lambda)\rho'(s,a) \log \frac{(1-\lambda)\rho'(s,a)}{(1-\lambda)\sum_{a'}\rho'(s,a')} \tag{21}$$

$$= \lambda\left(-\rho(s,a)\log\frac{\rho(s,a)}{\sum_{a'}\rho(s,a')}\right) + (1-\lambda)\left(-\rho'(s,a)\log\frac{\rho'(s,a)}{\sum_{a'}\rho'(s,a')}\right), \tag{22}$$

with equality if and only if $\pi_\rho \triangleq \rho(s,a)/\sum_{a'}\rho(s,a') = \rho'(s,a)/\sum_{a'}\rho'(s,a') \triangleq \pi_{\rho'}$. Summing both sides over all $s$ and $a$ shows that $\bar{H}(\lambda\rho + (1-\lambda)\rho') \geq \lambda\bar{H}(\rho) + (1-\lambda)\bar{H}(\rho')$ with equality if and only if $\pi_\rho = \pi_{\rho'}$. Applying Lemma 3.1 shows that equality in fact holds if and only if $\rho = \rho'$, so $\bar{H}$ is strictly concave.

Now, we turn to verifying the last two statements, which also follow from Lemma 3.1 and the definition of occupancy measures. First,

$$H(\pi) = \mathbb{E}_\pi[-\log\pi(a|s)] = -\sum_{s,a}\rho_\pi(s,a)\log\pi(a|s) \tag{23}$$

$$= -\sum_{s,a}\rho_\pi(s,a)\log\frac{\rho_\pi(s,a)}{\sum_{a'}\rho_\pi(s,a')} = \bar{H}(\rho_\pi), \tag{24}$$

and second,

$$\bar{H}(\rho) = -\sum_{s,a}\rho(s,a)\log\frac{\rho(s,a)}{\sum_{a'}\rho(s,a')} = -\sum_{s,a}\rho_{\pi_\rho}(s,a)\log\pi_\rho(a|s) \tag{25}$$

$$= \mathbb{E}_{\pi_\rho}[-\log\pi_\rho(a|s)] = H(\pi_\rho). \tag{26}$$

$\square$

*Proof of Proposition 3.1.*  This proof relies on properties of saddle points. For a reference, we refer the reader to Hiriart-Urruty and Lemaréchal [9, section VII.4].

Keeping $\mathcal{C} = \mathbb{R}^{\mathcal{S}\times\mathcal{A}}$, let $\tilde{c} \in \text{IRL}_\psi(\pi_E)$, $\tilde{\pi} \in \text{RL}(\tilde{c}) = \text{RL} \circ \text{IRL}_\psi(\pi_E)$, and

$$\pi_A \in \arg\min_\pi -H(\pi) + \psi^*(\rho_\pi - \rho_{\pi_E}) \tag{27}$$

$$= \arg\min_\pi\sup_{c\in\mathcal{C}} -H(\pi) - \psi(c) + \sum_{s,a}(\rho_\pi(s,a) - \rho_{\pi_E}(s,a))c(s,a) \tag{28}$$

We wish to show that $\pi_A = \tilde{\pi}$. To do this, let $\rho_A$ be the occupancy measure of $\pi_A$, let $\tilde{\rho}$ be the occupancy measure of $\tilde{\pi}$, and define $\bar{L}: \mathcal{D} \times \mathcal{C} \to \mathbb{R}$ by

$$\bar{L}(\rho, c) = -\bar{H}(\rho) - \psi(c) + \sum_{s,a}\rho(s,a)c(s,a) - \sum_{s,a}\rho_{\pi_E}(s,a)c(s,a). \tag{29}$$

The following relationships then hold, due to Lemma 3.1:

$$\rho_A \in \arg\min_{\rho\in\mathcal{D}}\sup_{c\in\mathcal{C}} \bar{L}(\rho, c), \tag{30}$$

$$\tilde{c} \in \arg\max_{c\in\mathcal{C}}\min_{\rho\in\mathcal{D}} \bar{L}(\rho, c), \tag{31}$$

$$\tilde{\rho} \in \arg\min_{\rho\in\mathcal{D}} \bar{L}(\rho, \tilde{c}). \tag{32}$$

(Recall that we can write Eq. (31) because we assumed the existence of a solution to the IRL problem Eq. (1).) Now $\mathcal{D}$ is compact and convex and $\mathcal{C}$ is convex; furthermore, due to convexity of $-\bar{H}$ and $\psi$, we also have that $\bar{L}(\cdot, c)$ is convex for all $c$, and that $\bar{L}(\rho, \cdot)$ is concave for all $\rho$, and hence:

$$\min_{\rho \in \mathcal{D}} \sup_{c \in \mathcal{C}} \bar{L}(\rho, c) = \max_{c \in \mathcal{C}} \min_{\rho \in \mathcal{D}} \bar{L}(\rho, c) \tag{33}$$

Consequently, from Eqs. (30) and (31), $(\rho_A, \tilde{c})$ is a saddle point of $\bar{L}$. In particular,

$$\rho_A \in \arg\min_{\rho \in \mathcal{D}} \bar{L}(\rho, \tilde{c}). \tag{34}$$

Because $\bar{L}(\cdot, c)$ is strictly convex for all $c$ (Lemma 3.2), Eqs. (32) and (34) imply $\rho_A = \tilde{\rho}$. Since policies corresponding to occupancy measures are unique (Lemma 3.1), $\pi_A = \tilde{\pi}$. □

## A.2 Proofs for Section 5

In Eq. (13) of Section 5, we described a cost regularizer $\psi_{\text{GA}}$, which leads to an imitation learning algorithm (15) that minimizes Jensen-Shannon divergence between occupancy measures (for clarity throughout, just as in Eq. (15), we will drop the $1 - \gamma$ normalization factor that converts occupancy measures to distributions). To justify our choice of $\psi_{\text{GA}}$, we show how to convert certain surrogate loss functions $\phi$, for binary classification of state-action pairs drawn from the occupancy measures $\rho_\pi$ and $\rho_{\pi_E}$, into cost function regularizers $\psi$, for which $\psi^*(\rho_\pi - \rho_{\pi_E})$ is the minimum expected risk $R_\phi(\rho_\pi, \rho_{\pi_E})$ for $\phi$:

$$R_\phi(\pi, \pi_E) = \sum_{s,a} \inf_{\gamma \in \mathbb{R}} \rho_\pi(s, a)\phi(\gamma) + \rho_{\pi_E}(s, a)\phi(-\gamma) \tag{35}$$

Specifically, we will restrict ourselves to strictly decreasing convex loss functions. Nguyen et al. [17] show a correspondence between minimum expected risks $R_\phi$ and symmetric $f$-divergences, of which Jensen-Shannon divergence is a special case. Our following construction, therefore, can generate any imitation learning algorithm that minimizes a symmetric $f$-divergence between occupancy measures, as long as that $f$-divergence is induced by a strictly decreasing convex surrogate $\phi$.

**Proposition A.1.** *Suppose* $\phi : \mathbb{R} \to \mathbb{R}$ *is a strictly decreasing convex function. Let $T$ be the range of* $-\phi$, *and define* $g_\phi : \mathbb{R} \to \bar{\mathbb{R}}$ *and* $\psi_\phi : \mathbb{R}^{\mathcal{S} \times \mathcal{A}} \to \bar{\mathbb{R}}$ *by:*

$$g_\phi(x) = \begin{cases} -x + \phi(-\phi^{-1}(-x)) & \text{if } x \in T \\ +\infty & \text{otherwise} \end{cases}$$

$$\psi_\phi(c) = \begin{cases} \displaystyle\sum_{s,a} \rho_{\pi_E}(s, a) g_\phi(c(s, a)) & \text{if } c(s, a) \in T \text{ for all } s, a \\ +\infty & \text{otherwise} \end{cases} \tag{36}$$

*Then,* $\psi_\phi$ *is closed, proper, and convex, and* $\text{RL} \circ \text{IRL}_{\psi_\phi}(\pi_E) = \arg\min_\pi -H(\pi) - R_\phi(\rho_\pi, \rho_{\pi_E})$.

*Proof.* To verify the first claim, it suffices to check that $g_\phi(x) = -x + \phi(-\phi^{-1}(-x))$ is closed, proper, and convex. Convexity follows from the fact that $x \mapsto \phi(-\phi^{-1}(-x))$ is convex, because it is a concave function followed by a nonincreasing convex function. Furthermore, because $T$ is nonempty, $g_\phi$ is proper. To show that $g_\phi$ is closed, note that because $\phi$ is strictly decreasing and convex, the range of $\phi$ is either all of $\mathbb{R}$ or an open interval $(b, \infty)$ for some $b \in \mathbb{R}$. If the range of $\phi$ is $\mathbb{R}$, then $g_\phi$ is finite everywhere and is therefore closed. On the other hand, if the range of $\phi$ is $(b, \infty)$, then $\phi(x) \to b$ as $x \to \infty$, and $\phi(x) \to \infty$ as $x \to -\infty$. Thus, as $x \to b$, $\phi^{-1}(-x) \to \infty$, so $\phi(-\phi^{-1}(-x)) \to \infty$ too, implying that $g_\phi(x) \to \infty$ as $x \to b$, which means $g_\phi$ is closed.

Now, we verify the second claim. By Proposition 3.1, all we need to check is that $-R_\phi(\rho_\pi, \rho_{\pi_E}) = \psi_\phi^*(\rho_\pi - \rho_{\pi_E})$:

$$\psi_\phi^*(\rho_\pi - \rho_{\pi_E}) = \sup_{c \in \mathcal{C}} \sum_{s,a} (\rho_\pi(s,a) - \rho_{\pi_E}(s,a))c(s,a) - \sum_{s,a} \rho_{\pi_E}(s,a)g_\phi(c(s,a)) \tag{37}$$

$$= \sum_{s,a} \sup_{c \in T} (\rho_\pi(s,a) - \rho_{\pi_E}(s,a))c - \rho_{\pi_E}(s,a)[-c + \phi(-\phi^{-1}(-c))] \tag{38}$$

$$= \sum_{s,a} \sup_{c \in T} \rho_\pi(s,a)c - \rho_{\pi_E}(s,a)\phi(-\phi^{-1}(-c)) \tag{39}$$

$$= \sum_{s,a} \sup_{\gamma \in \mathbb{R}} \rho_\pi(s,a)(-\phi(\gamma)) - \rho_{\pi_E}(s,a)\phi(-\phi^{-1}(\phi(\gamma))) \tag{40}$$

$$= \sum_{s,a} \sup_{\gamma \in \mathbb{R}} \rho_\pi(s,a)(-\phi(\gamma)) - \rho_{\pi_E}(s,a)\phi(-\gamma) \tag{41}$$

$$= -R_\phi(\rho_\pi, \rho_{\pi_E}) \tag{42}$$

where we made the change of variables $c \to -\phi(\gamma)$, justified because $T$ is the range of $-\phi$. $\square$

Having showed how to construct a cost function regularizer $\psi_\phi$ from $\phi$, we obtain, as a corollary, a cost function regularizer for the logistic loss, whose optimal expected risk is, up to a constant, the Jensen-Shannon divergence.

**Corollary A.1.1.** *The cost regularizer* (13)

$$\psi_{GA}(c) \triangleq \begin{cases} \mathbb{E}_{\pi_E}[g(c(s,a))] & \text{if } c < 0 \\ +\infty & \text{otherwise} \end{cases} \quad \text{where} \quad g(x) = \begin{cases} -x - \log(1 - e^x) & \text{if } x < 0 \\ +\infty & \text{otherwise} \end{cases}$$

*satisfies*

$$\psi_{GA}^*(\rho_\pi - \rho_{\pi_E}) = \sup_{D \in (0,1)^{S \times A}} \mathbb{E}_\pi[\log(D(s,a))] + \mathbb{E}_{\pi_E}[\log(1 - D(s,a))]. \tag{43}$$

*Proof.* Using the logistic loss $\phi(x) = \log(1 + e^{-x})$, we see that Eq. (36) reduces to the claimed $\psi_{GA}$. Applying Proposition A.1, we get

$$\psi_{GA}^*(\rho_\pi - \rho_{\pi_E}) = -R_\phi(\rho_\pi, \rho_{\pi_E}) \tag{44}$$

$$= \sum_{s,a} \sup_{\gamma \in \mathbb{R}} \rho_\pi(s,a) \log\left(\frac{1}{1 + e^{-\gamma}}\right) + \rho_{\pi_E}(s,a) \log\left(\frac{1}{1 + e^\gamma}\right) \tag{45}$$

$$= \sum_{s,a} \sup_{\gamma \in \mathbb{R}} \rho_\pi(s,a) \log\left(\frac{1}{1 + e^{-\gamma}}\right) + \rho_{\pi_E}(s,a) \log\left(1 - \frac{1}{1 + e^{-\gamma}}\right) \tag{46}$$

$$= \sum_{s,a} \sup_{\gamma \in \mathbb{R}} \rho_\pi(s,a) \log(\sigma(\gamma)) + \rho_{\pi_E}(s,a) \log(1 - \sigma(\gamma)), \tag{47}$$

where $\sigma(x) = 1/(1 + e^{-x})$ is the sigmoid function. Because the range of $\sigma$ is $(0, 1)$, we can write

$$\psi_{GA}^*(\rho_\pi - \rho_{\pi_E}) = \sum_{s,a} \sup_{d \in (0,1)} \rho_\pi(s,a) \log d + \rho_{\pi_E}(s,a) \log(1 - d) \tag{48}$$

$$= \sup_{D \in (0,1)^{S \times A}} \sum_{s,a} \rho_\pi(s,a) \log(D(s,a)) + \rho_{\pi_E}(s,a) \log(1 - D(s,a)), \tag{49}$$

which is the desired expression. $\square$

We conclude with a policy gradient formula for causal entropy.

**Lemma A.1.** *The causal entropy gradient is given by*

$$\nabla_\theta \mathbb{E}_{\pi_\theta}[-\log \pi_\theta(a|s)] = \mathbb{E}_{\pi_\theta}[\nabla_\theta \log \pi_\theta(a|s)Q_{\log}(s,a)],$$
$$\text{where} \quad Q_{\log}(\bar{s}, \bar{a}) = \mathbb{E}_{\pi_\theta}[-\log \pi_\theta(a|s) \mid s_0 = \bar{s}, a_0 = \bar{a}]. \tag{50}$$

*Proof.* For an occupancy measure $\rho(s, a)$, define $\rho(s) = \sum_a \rho(s, a)$. Next,

$$\nabla_\theta \mathbb{E}_{\pi_\theta}[-\log \pi_\theta(a|s)] = -\nabla_\theta \sum_{s,a} \rho_{\pi_\theta}(s, a) \log \pi_\theta(a|s)$$

$$= -\sum_{s,a} (\nabla_\theta \rho_{\pi_\theta}(s, a)) \log \pi_\theta(a|s) - \sum_s \rho_{\pi_\theta}(s) \sum_a \pi_\theta(a|s) \nabla_\theta \log \pi_\theta(a|s)$$

$$= -\sum_{s,a} (\nabla_\theta \rho_{\pi_\theta}(s, a)) \log \pi_\theta(a|s) - \sum_s \rho_{\pi_\theta}(s) \sum_a \nabla_\theta \pi_\theta(a|s)$$

The second term vanishes, because $\sum_a \nabla_\theta \pi_\theta(a|s) = \nabla_\theta \sum_a \pi_\theta(a|s) = \nabla_\theta 1 = 0$. We are left with

$$\nabla_\theta \mathbb{E}_{\pi_\theta}[-\log \pi_\theta(a|s)] = \sum_{s,a} (\nabla_\theta \rho_{\pi_\theta}(s, a))(-\log \pi_\theta(a|s)),$$

which is the policy gradient for RL with the fixed cost function $c_{\log}(s, a) \triangleq -\log \pi_\theta(a|s)$. The resulting formula is given by the standard policy gradient formula for $c_{\log}$. $\qquad \square$

## B  Environments and detailed results

The environments we used for our experiments are from the OpenAI Gym [5]. The names and version numbers of these environments are listed in Table 1, which also lists dimension or cardinality of their observation and action spaces (numbers marked "continuous" indicate dimension for a continuous space, and numbers marked "discrete" indicate cardinality for a finite space).

As outlined in Section 6, our experiment pipeline for a single environment consists of the following steps: (1) training an expert with TRPO on the true cost function, (2) sampling a dataset of trajectories from the expert, and (3) running imitation learning algorithms on that dataset. (Note that the imitation learning algorithms, over multiple reruns, are given the same datasets.) The performance of the TRPO-trained experts and the performance of random policies are listed in Table 1.

Table 1: Environments

| Task | Observation space | Action space | Random policy performance | Expert performance |
|---|---|---|---|---|
| Cartpole-v0 | 4 (continuous) | 2 (discrete) | $18.64 \pm 7.45$ | $200.00 \pm 0.00$ |
| Acrobot-v0 | 4 (continuous) | 3 (discrete) | $-200.00 \pm 0.00$ | $-75.25 \pm 10.94$ |
| MountainCar-v0 | 2 (continuous) | 3 (discrete) | $-200.00 \pm 0.00$ | $-98.75 \pm 8.71$ |
| Reacher-v1 | 11 (continuous) | 2 (continuous) | $-43.21 \pm 4.32$ | $-4.09 \pm 1.70$ |
| HalfCheetah-v1 | 17 (continuous) | 6 (continuous) | $-282.43 \pm 79.53$ | $4463.46 \pm 105.83$ |
| Hopper-v1 | 11 (continuous) | 3 (continuous) | $14.47 \pm 7.96$ | $3571.38 \pm 184.20$ |
| Walker-v1 | 17 (continuous) | 6 (continuous) | $0.57 \pm 4.59$ | $6717.08 \pm 845.62$ |
| Ant-v1 | 111 (continuous) | 8 (continuous) | $-69.68 \pm 111.10$ | $4228.37 \pm 424.16$ |
| Humanoid-v1 | 376 (continuous) | 17 (continuous) | $122.87 \pm 35.11$ | $9575.40 \pm 1750.80$ |

To generate the datasets, we subsampled the expert trajectories for the different environments at various timestep intervals: 10 timesteps between samples for Cartpole, 5 for Mountain Car and Acrobot, 1 for Reacher, and 20 for Hopper, Walker, Ant, HalfCheetah, and Humanoid. This both made the tasks harder and made the amount of data given to the algorithms approximately comparable over the various tasks, as the average trajectory lengths of the various environments differ vastly from each other.

The amount of environment interaction used for FEM, GTAL, and GAIL is shown in Table 2. To reduce gradient variance for these three algorithms, we also fit value functions, with the same neural network architecture as the policies, and employed generalized advantage estimation [25] (with $\gamma = .995$ and $\lambda = .97$). The exact experimental results are listed in Table 3. Means and standard deviations are computed over a number of runs with different random seeds: 7 runs for Cartpole, Acrobot, Mountain Car, and Reacher; 5 runs for HalfCheetah, Hopper, Walker, and Ant; 1 run for Humanoid. The policy learned from each run is assessed by its average performance over 50 rollouts.

Table 2: Parameters for FEM, GTAL, and GAIL

| Task | Training iterations | State-action pairs per iteration |
|------|---------------------|----------------------------------|
| Cartpole | 300 | 5000 |
| Mountain Car | 300 | 5000 |
| Acrobot | 300 | 5000 |
| Reacher | 200 | 50000 |
| HalfCheetah | 500 | 50000 |
| Hopper | 500 | 50000 |
| Walker | 500 | 50000 |
| Ant | 500 | 50000 |
| Humanoid | 1500 | 50000 |

Table 3: Learned policy performance

| Task | Dataset size | Behavioral cloning | FEM | GTAL | GAIL (ours) |
|------|--------------|--------------------|-----|------|-------------|
| Cartpole | 1 | $71.94 \pm 23.94$ | $200.00 \pm 0.00$ | $200.00 \pm 0.00$ | $200.00 \pm 0.00$ |
| | 4 | $168.98 \pm 48.67$ | $200.00 \pm 0.00$ | $200.00 \pm 0.00$ | $200.00 \pm 0.00$ |
| | 7 | $188.60 \pm 20.54$ | $200.00 \pm 0.00$ | $199.94 \pm 0.14$ | $200.00 \pm 0.00$ |
| | 10 | $177.19 \pm 46.85$ | $199.75 \pm 0.62$ | $200.00 \pm 0.00$ | $200.00 \pm 0.00$ |
| Acrobot | 1 | $-130.60 \pm 36.10$ | $-133.32 \pm 57.78$ | $-81.35 \pm 3.30$ | $-77.28 \pm 4.00$ |
| | 4 | $-93.20 \pm 9.64$ | $-94.21 \pm 43.23$ | $-94.80 \pm 43.23$ | $-83.12 \pm 3.49$ |
| | 7 | $-96.92 \pm 6.80$ | $-94.99 \pm 43.13$ | $-95.72 \pm 42.88$ | $-82.56 \pm 4.44$ |
| | 10 | $-95.10 \pm 4.52$ | $-77.22 \pm 3.75$ | $-94.32 \pm 43.38$ | $-78.91 \pm 1.28$ |
| Mountain Car | 1 | $-136.75 \pm 6.44$ | $-100.98 \pm 3.23$ | $-115.44 \pm 34.61$ | $-101.55 \pm 2.14$ |
| | 4 | $-133.25 \pm 4.27$ | $-99.29 \pm 1.76$ | $-143.58 \pm 48.96$ | $-101.35 \pm 1.18$ |
| | 7 | $-127.34 \pm 9.08$ | $-100.65 \pm 1.49$ | $-128.96 \pm 44.99$ | $-99.90 \pm 0.79$ |
| | 10 | $-123.14 \pm 7.31$ | $-100.48 \pm 0.97$ | $-120.00 \pm 34.29$ | $-100.83 \pm 2.81$ |
| HalfCheetah | 4 | $-319.88 \pm 306.80$ | $338.97 \pm 468.81$ | $432.97 \pm 769.97$ | $4274.05 \pm 251.75$ |
| | 11 | $184.78 \pm 440.31$ | $71.83 \pm 511.65$ | $273.68 \pm 417.29$ | $4498.79 \pm 226.55$ |
| | 18 | $2344.70 \pm 1313.61$ | $-165.07 \pm 482.60$ | $739.77 \pm 929.13$ | $4729.51 \pm 124.86$ |
| | 25 | $2849.87 \pm 954.66$ | $242.13 \pm 247.16$ | $-95.08 \pm 520.46$ | $4823.48 \pm 46.40$ |
| Hopper | 4 | $394.99 \pm 25.24$ | $3460.88 \pm 82.30$ | $2842.52 \pm 915.60$ | $3604.94 \pm 18.85$ |
| | 11 | $1503.65 \pm 910.70$ | $3509.89 \pm 113.57$ | $2758.53 \pm 668.62$ | $3607.44 \pm 18.07$ |
| | 18 | $1928.39 \pm 1036.67$ | $3519.44 \pm 94.34$ | $2591.56 \pm 858.33$ | $3631.70 \pm 14.09$ |
| | 25 | $2022.83 \pm 665.59$ | $3443.99 \pm 114.51$ | $3043.17 \pm 462.35$ | $3615.54 \pm 11.51$ |
| Walker | 4 | $548.31 \pm 357.68$ | $4449.50 \pm 805.64$ | $4379.85 \pm 1103.55$ | $6675.04 \pm 348.72$ |
| | 11 | $2534.97 \pm 1508.82$ | $3784.84 \pm 391.31$ | $4835.57 \pm 518.24$ | $6884.47 \pm 169.34$ |
| | 18 | $2846.40 \pm 2033.15$ | $3795.22 \pm 275.70$ | $4433.71 \pm 784.88$ | $6947.50 \pm 146.28$ |
| | 25 | $3348.29 \pm 1186.94$ | $4077.99 \pm 414.05$ | $4888.72 \pm 423.27$ | $7027.03 \pm 76.39$ |
| Ant | 4 | $1384.42 \pm 212.60$ | $-4510.92 \pm 2328.03$ | $-4042.96 \pm 1998.47$ | $3233.16 \pm 310.87$ |
| | 11 | $2622.63 \pm 309.01$ | $-3550.70 \pm 575.79$ | $-4240.46 \pm 1704.07$ | $3894.09 \pm 324.45$ |
| | 18 | $3048.75 \pm 150.00$ | $-4586.88 \pm 2001.42$ | $-4949.60 \pm 1861.02$ | $3684.49 \pm 285.52$ |
| | 25 | $3598.54 \pm 578.18$ | $-5457.76 \pm 1389.80$ | $-5404.55 \pm 1054.70$ | $4057.52 \pm 393.90$ |
| Humanoid | 80 | 1397.06 | 5093.12 | 5096.43 | 10200.73 |
| | 160 | 3655.14 | 5120.52 | 5412.47 | 10119.80 |
| | 240 | 5660.53 | 5192.34 | 5145.94 | 10361.94 |

| Task | Dataset size | Behavioral cloning | GAIL ($\lambda = 0$) | GAIL ($\lambda = 10^{-3}$) | GAIL ($\lambda = 10^{-2}$) |
|------|--------------|--------------------|----------------------|---------------------------|---------------------------|
| Reacher | 4 | $-10.97 \pm 2.49$ | $-67.23 \pm 34.70$ | $-32.37 \pm 17.57$ | $-46.72 \pm 49.52$ |
| | 11 | $-6.23 \pm 0.69$ | $-6.06 \pm 0.89$ | $-6.61 \pm 1.30$ | $-9.23 \pm 7.80$ |
| | 18 | $-4.76 \pm 0.32$ | $-8.25 \pm 5.77$ | $-5.66 \pm 0.57$ | $-5.04 \pm 0.35$ |