[Reviews · NeurIPS 2016]

Reviewer 1

Summary

The paper proposes a new way of imitation learning. It aims at the generality and of inverse reinforcement followed by a reinforcement learning step without actually doing both steps. By employing the ideas of generative adversarial training they learn a discriminator and a generator for policies imitating reference trajectories. The paper provides a framework for expressing the different algorithms in a unified way. Experiments show its effectiveness over simple supervised behavioral cloning but also recently proposed methods.

Qualitative Assessment

The paper is well written and provides a nice theoretical framework for deriving their work, and embedding and comparing it to previous work. The paper is dense and contains a lot of information. Although, I am not quite sure all the math was needed for the message of the paper. It this way the results and discussion are a bit short. There is a weakness in the approach w.r.t. cloning, which is shortly mentioned in the discussion. It is the number of environment interactions needed. It would be good to give the exact numbers for comparison in the result section. Language issues: l 52: with respect TO a l 202: stray to far FROM ? l 211: wields -> yields l 237: We are now treat ...

Confidence in this Review

2-Confident (read it all; understood it all reasonably well)


Reviewer 2

Summary

The manuscript considers the problem of imitation learning, i.e. learning to do a task by observing expert data (but never the underlying cost function). They develop a new approach that is cast in the "generative adversarial" framework. They test the approach on 9 simulated continuous RL tasks and show that on most of these the new approach outperforms reasonable baselines.

Qualitative Assessment

The manuscript presents a novel approach to imitation learning. It draw and prove a series of interesting connections to the existing literature and frames the approach in terms of generative adversarial learning. The experiments demonstrate compelling performance of the approach against reasonable baselines. The major limitation of the approach in its current form, which the manuscript could stress a bit more in the introduction, is that the approach still requires full access to the system that it is trying to control (even if it never sees rewards and has a fixed amount of expert data), and a lot of data from this system [approximately as much data as is required to learn the expert policy in the first place]. This point could be emphasized with a figure or table as well -- currently this limitation is a bit obscured in the discussion. The approach is interesting enough as it is, and making the limitation clear will only serve to strengthen the clarity and completeness of the paper. It would be nice, as suggested in the manuscript, if this limitation could be addressed in part by initializing with behavioural cloning. Showing that this is significantly the case would further strengthen the paper. I feel that the paper could be made more clear by introducing the proposed algorithm earlier and generally simplifying the presentation. The effort to situate the approach within the literature is appreciated of course, but it takes a very long time to get to what is at core a simple idea that is intuitively intelligible without much background. By introducing the basic algorithm (i.e. the essential ideas in Section 5.1) earlier, the reader could be led through the literature in a more directed fashion. In general, clarity could be improved by moving some pieces of the less crucial math to the supplement and introducing a simple figure or two that help the reader appreciate the basic approach better [e.g. a diagram of the basic setup and illustration of, e.g. the proposed function in 16]. As it stands, I felt that the paper made things more mysterious than they needed to be and required a second read-through to really appreciate why certain tools are developed. As well, because of the lengthy development of the background, the experimental section felt light on details that could help others replicate the work. Minor Issues: Causal entropy regularization is discussed throughout the background and development, but then it's only ever used in the reacher task [line 300]. Perhaps it would be easier to have it play less of a role in the theoretical development if it's just dropped from the experiments? Figure 1: "Simplex" is probably a bad label, or at least mismatched with what is usually used in the text? Grammar/Spelling issues on lines: 148 -- "with expert" 237 -- "we are now treat" 258 -- "an empirical expectations" 277 -- "performance respect"

Confidence in this Review

3-Expert (read the paper in detail, know the area, quite certain of my opinion)


Reviewer 3

Summary

In this paper the authors proposed a model-free imitation learning that 1) obtains significant performance gains over existing model-free methods in imitating learning/behavioral cloning, and 2) bypasses the indirect nature of inverse reinforcement learning where a cost function is first learned with expert's data and an optimal policy is obtained by RL afterwards.

Qualitative Assessment

In general the paper has a good theoretical contribution to IRL and apprenticeship learning. By drawing the equivalence between policy improvement (wrt to expert's policy) and occupation measure matching via strong duality, the authors proposed a novel way of solving the min-max optimization in apprenticeship learning, and they showed that the newly proposed formulation includes traditional approaches from Abbeel and Ng. Second, by leveraging the recently popular technique of general adversarial networks, they proposed a practical algorithm with a choice of regularizer \psi that resembles JS divergence in adversarial imitation learning. The concepts are well-explained in this paper, and they are quite well written. I found this paper contains some solid contributions, however there are several comments to this paper: 1) Why do you need a proof for Lemma 8.1? Seems it is a fairly standard result in policy gradient methods. 2) The current occupation measure definition has a (1-\gamma) factor missing (i.e. the current definition of \rho is an occupation measure but not a probability distribution). Also, similar analysis of Prop 3.1 can be found in Section 6.9 of Puterman's book: Markov Decision Processes: Discrete Stochastic Dynamic Programming. 3) The equivalence between the optimization of occupation measures and optimization of policy is well-known in the dual MDP setup. For example the authors may find some technical results in Altman's book: Constrained MDP to streamline the analysis of the saddle point theorems. 4) Can you state the standard assumption on the constraint qualifications (i.e. Slater's condition) when strong duality is used in convex optimization? 5) The analysis in line 22 to 25 is quite standard and can be found in convex analysis book such as Boyd's book. 6) It also seems that the importance of lemmas and corollaries in Section 3 is swapped (i.e. the proof of lemma 3.1 is in appendix, where the proof of corollary 3.2.1 is in the main paper). 7) The proof of Corollary 3.2.1, which is the main technical result in Section 3 (or even the paper) is quite sloppy. For example set D needs to be compact (which is clearly true), and again Slater's condition is required in strong duality arguments. Please improve the analysis to make it technically sound and rigorous. 8) In Equation 12 when the gradients and max are swapped, please check the conditions in envelop theorem (see Envelope Theorems for Arbitrary Choice Sets by Milgrom). 9) Please state all the assumptions in Proposition 8.1 instead of citing it from a separate paper. It makes readers very difficult to parse. Also, please put the result in the main paper. What are the underlying motivations of defining the functions \gamma and \phi when they are not appearing in the main paper? Also what is the functional domain of \gamma?

Confidence in this Review

3-Expert (read the paper in detail, know the area, quite certain of my opinion)


Reviewer 4

Summary

The paper considers imitation learning. In particular, it tries to combine the strengths of behavioral cloning with the strengths inverse reinforcement learning while remaining data-efficient. To do so, the paper establishes theoretical framework based on occupancy measures and Fenchel conjugates. Theoretical properties of the framework are established or cited to justify the framework, and it is shown that many approaches to imitation learning can indeed be cast into this framework. Once the framework is established, an instance of imitation learning is introduced. The respective learning algorithm is derived and shown to be similar to the learning algorithm for Generative Adversarial Networks. The algorithm is evaluated on multiple tasks of differing cmoplexity and shown to outperform previous approaches.

Qualitative Assessment

The paper provides a very interesting perspective onto imitation learning. Casting many previous approaches into one unified framework can give rise to new insights and extensions, such as, but not limited to, the provided Generative Adversarial Imitation Learning (GAIL) algorithm. Parts of the framework are not entirely new, occupancy functions have been used previously (as stated by the authors). Nevertheless, noteworthy new theoretical statements based on analysis from a (convex) optimization point of few as well as connections to previous approaches are established, the main result being proposition 3.2. My main critique concerns the structure of the paper: Title and abstract are dominated by GAIL, one particular instantiation of their framework. The paper, however, spends five pages on a thorough theoretical analysis before introducing GAIL on one page and evaluating it on another page. While providing interesting insights, the more theoretical parts, i.e., sections 2 to 4, could be made more clear and streamlined in some parts: Right in the introduction of section 2, the analysis is restricted to finite state and action space, due to reduced "technical machinery". From the paper, it is unclear whether the theory transfers entirely to continuous spaces. Also, it is unclear which parts of the framework are truly original contributions of the authors, as they are intertwined with definitions and theorems from other papers, which are not always cited properly. As an example, Eq. (3) and Prop. 3.2 are novel contributions, while the text in between seems to state previous work without proper citations (except for Prop. 3.1). At first glance, it is unclear why the proof of Corr. 3.2.1 is the only proof that makes it into the paper. From my understanding is that the proof requires lemmata 3.1 and 3.2, which themselves are interesting properties of the framework. In short: A stricter definitions-properties-consequences structure of sections 2 to 4 would speed up the understanding of the theory (and as a byproduct probably shorten the respective sections). A note on the supplementary material: The readability would be improved greatly if (i) the order of the proofs matched the order of the corresponding theorems in the paper, and (ii) the theorem to be proven is restated so that the reader does not have to constantly flip pages between theorem and proof. Because of the (understandably) strong focus on developing the theoretical framework, the experimental section becomes too short. The experimental setup is described insufficiently. The data set generation is not described properly (in particular not quantitatively, which prohibits reproduction of the results). for behavioral cloning only a batch size is given. Relevant results are moved to the supplementary material and the method of evaluation becomes more coarse the more interesting the problem is. The results are not evaluated thoroughly, leading to very vague statements such as "nearly always dominating all the baselines". In one experiments, GAIL underperforms, this is not analyzed properly (only a fix is introduced, which improves, but does not solve the underperformance). Figure (1g) could also use more explanation---why is it not monotonic like all other curves? That said, there are some small glitches that can easily be fixed: - line 68: You introduce this isomorphism to a real vector space. This would be much more evident if the dimension was not a set, but the cardinality of a set (standard notation) and you simply referred to it as a "cost matrix", which it is essentially. - line 77: will --> we will - line 104: broken reference, just refer to the proposition instead. - lines 127/128: Put that equation into its own environment for readability. - line 228: For layout consistency, state this as a theorem as you do in section 3, and then refer to a proof in the appendix. -line 237: Delete "are". As a summary: The paper can be improved in many ways, the most important being a shift of focus towards theory and a few more words on the applicability of the theory in continuous space. The theoretical work is a very interesting contribution, which may be very valuable for other researchers at NIPS. The application to GAIL is also interesting, but the evaluation is too coarse to put the main focus on GAIL.

Confidence in this Review

2-Confident (read it all; understood it all reasonably well)


Reviewer 5

Summary

The paper is concerned with methods trying to imitate expert behavior via imitation learning. It presents both a theoretical investigation into connections between existing imitation learning (or more concretely, apprenticeship learning) algorithms and a new algorithm for imitation learning via a method akin to the training of generative adversarial networks. Extensive experiments are conducted on a variety of benchmarks showing the efficacy of the derived algorithm.

Qualitative Assessment

First off, this is a clearly written and well thought through paper that absolutely deserves to be published at NIPS. The derivation of inverse reinforcement learning as the dual of learning an occupancy measure in the state-action space is clear, concise, and allows the authors to highlight connections between different, existing, IRL algorithms. These were interesting to discover for me as a reader and the proofs all seem sound. The newly derived adversarial imitation learning method, which draws on a connection to generative adversarial networks (GANs) (formulating the problem of learning an occupancy measure as a classification problem), is interesting and seems less limited than the previously existing state of the art (in terms of applicability to a wide range of different tasks). I suspect that we will see several new advancements based on this new algorithm as it naturally invites for further application of existing deep learning machinery to the problem. One argument that could be made against the paper is that it is fairly dense, packing a lot of material into 8 pages, and thus fails to be completely self-contained (i.e. the steps required to optimize the policy via TRPO are not well explained). While true, I believe this generally to be tolerable in this case as the paper is well written and explains all necessary concepts in enough detail for the reader to follow. However, I do believe that an explanation of the exact algorithm, including the TRPO steps would be useful for reproducing the paper, and aiding the interested reading in following each step. I thus would encourage the authors to add such a description to the supplementary material. The only real issue I have with the paper is the fact that the authors seem to gloss over the problems that using an GAN like procedure can introduce. It is well known that GANs can be difficult to train and I suspect that the usage in the presented algorithm makes no exception to that rule. To that end it would be interesting to understand which (if any) countermeasures the authors took to stabilize training. Touching on the same subject, the objective maximized by the discriminator (Equation (19)) results in rewards going towards -inf for the policy (in cases where it does not currently match the expert behaviour), this could result in very large variances in the likelihood ratio estimator used for optimizing the policy parameters with TRPO. I suspect the authors thus implemented some counter measures in their actual implementation, ensuring numerical stability, that would be nice to know. Furthermore, while the evaluation is extensive, covering a large amount of different domains, the effect of random initialization of the networks seems to be neglected. To that end it would be valuable to see error bars for Figures 1-2 and a note on how stable the presented procedure is with respect to initialization and hyperparameter choices. Apart from that I only have a few minor comments: - While the paper is well written articles are sometimes missing or wrongly placed, it would be a shame if these small mistakes make it into the paper, here are the ones I found: Abstract: - "without interaction with expert or access to reinforcement signal" -> "without interaction with expert or access to a reinforcement signal" Introduction: - "Given that learner’s true goal often is" -> "Given that the learner’s true goal often is" - "a technique from deep learning community that has" -> "a technique from the deep learning community that has" - "will in fact find that \psi plays a central role in our discussion, not as a nuisance in our analysis." -> "we will in fact find that \psi plays a central role in our discussion, and not only plays the role of a nuisance in our analysis." Section 5: - "We are now treat the causal entropy" -> "We now treat the causal entropy" - "in contrast with linear apprenticeship" -> "in contrast to linear apprenticeship" - "adaptation of generative adversarial training principle" -> "adaptation of the generative adversarial training principle" - "addressed the capability of \psi_{GA} of exactly matching occupancy measures" -> "addressed the capability of \psi_{GA} to exactly match occupancy measures" Section 5.1: - "denote an empirical expectations over" -> "denote an empirical expectation over" Section 6: - "evaluate the imitation performance respect to sample" -> "evaluate the imitation performance with respect to sample" - In Section 6 (and I believe in other places as well) there is a forward reference to the supplementary (as Section 9). It would be good to explicitely write "Section 9 in the supplementary" (and perhaps name the Sections in the supplementary alphabetically) - In Section 5.1 you write that you use the ADAM learning rule but in fact never explicitely mention that you train on mini batches using stochastic gradient descent. Of course using ADAM implies this but it would be better to add it to the sentence.

Confidence in this Review

3-Expert (read the paper in detail, know the area, quite certain of my opinion)